# Differences in Insomnia Symptoms between Immigrants and Non-Immigrants in Switzerland attributed to Emotional Distress: Analysis of the Swiss Health Survey

**DOI:** 10.3390/ijerph16020289

**Published:** 2019-01-21

**Authors:** Andres R. Schneeberger, Azizi Seixas, Nina Schweinfurth, Undine E. Lang, Christian Cajochen, Donald A. Bux, Shannique Richards, Girardin Jean-Louis, Christian G. Huber

**Affiliations:** 1Psychiatrische Dienste Graubuenden (PDGR), Loestrasse 220, 7000 Chur, Switzerland; donald.bux@pdgr.ch; 2Department of Psychiatry and Behavioral Sciences, Albert Einstein College of Medicine (AECOM), 3331 Bainbridge Avenue, Bronx, NY 10467, USA; 3Psychiatrische Universitaetsklinik (PUK), Klinik für Psychiatrie, Psychotherapie und Psychosomatik, Universitaet Zuerich, Lenggstrasse 31, 8032 Zurich, Switzerland; 4Center for Healthful Behavior Change, Department of Population Health, NYU School of Medicine, Translational Research Building, 227 East 30th Street, Floor 7, New York, NY 10016, USA; Azizi.Seixas@nyumc.org (A.S.); shannique.richards@nyulangone.org (S.R.); Girardin.Jean-Louis@nyulangone.org (G.J.-L.); 5Universitaere Psychiatrische Kliniken Basel, Universitaet Basel, Switzerland (UPK), Wilhelm Klein-Strasse 27, 4012 Basel, Switzerland; Nina.Schweinfurth@upkbs.ch (N.S.); undine.lang@upkbs.ch (U.E.L.); Christian.Cajochen@upkbs.ch (C.C.); christian.huber@upkbs.ch (C.G.H.)

**Keywords:** German, Italian, Portuguese, trouble falling asleep, early morning awakening, restless sleep

## Abstract

Migration can be a stressful experience and may lead to poor health and behavioral changes. The immigrant population in Switzerland is disproportionately burdened by several negative health outcomes, chief among these is mental health issues. The aim of the study was to investigate whether sleep disturbances are more prevalent among immigrants compared to non-immigrants and whether emotional distress might explain sleep differences. Based on the Swiss Health Survey 2012 dataset, we analyzed the data of 17,968 people, of which 3406 respondents were immigrants. We examined variables including insomnia symptoms, emotional distress and clinical and socio-demographic data using unadjusted and adjusted generalized linear models. Compared to non-immigrants, immigrants suffer significantly more often from insomnia symptoms. Immigrants also endured higher levels of emotional distress. Higher values of emotional distress are related to other symptoms of sleep disorders. Immigrants with emotional distress were at significant risk of sleep disturbances. Sleep disparities between immigrants and non-immigrants may be influenced by emotional distress. Migration health care should address emotional distress, a more proximal and modifiable factor, as a possible cause of insomnia symptoms in immigrants.

## 1. Introduction

In 2015, a total of 2.44 billion people, or 32.9% of the entire world population, were immigrants [1]. There is an increasing trend of migration into developed regions. The United Nations described 2014 as the year with the highest level of forced migration on record. As of 2015, one out of 122 people worldwide was a refugee or was seeking asylum [2]. A total of 4.7 million people immigrated to a European Union member state during 2015 [3]. In 2014, 35.4% of the permanent resident population in Switzerland had a migration background (immigrants and children of immigrants). Of people with a migration background, 80% were immigrants themselves, whereas one fifth were born in Switzerland [4]. The majority of Switzerland’s foreign resident population is of European origin, most of whom are nationals of the European Union or a European Free Trade Association member state. The largest groups of migrants are Italian (15.3%), German (14.9%), Portuguese (13.1%) and French (5.8%) [5]. 

Health information on migrants’ health and their access to health services is scarce and contradictory. The frequency of adverse health outcomes among migrants is higher than that seen in the host population [6]. Analyses of Latino immigrants in the United States of America (U.S.) revealed more negative health outcomes such as depression, diabetes, and obesity than in the non-immigrant population [7]. Bischoff and Wanner [8] showed that the self-reported health of immigrants in Switzerland was inferior to that of the non-immigrant Swiss population as indexed by a history of symptoms of chronic conditions, lack of mental well-being, lack of a sense of mastery, physical inactivity; and obesity. This difference could not be explained by differences regarding access to mental health care, which was the same for both for immigrants and non-immigrants according to Swiss reports on the migrant population. Previous studies in Europe identified immigration as a health and sleep-relevant stress factor [9,10]. Similarly, Voss and Tuin [10] studied women who had integrated more into the host society and found that they slept worse than less-integrated women, suggesting that non-integration may serve a protective function. An Australian study analyzed the impact of acculturative stress on depression and sleep, showing that despite long durations of stay, health disparities persisted [11].

Prior studies have offered initial evidence that some psychosocial factors—such as education level, income, unemployment, and access to social resources—among immigrants or minority populations may explain the development and persistence of disturbed sleep [12,13]. However, there is little population-based research on the impact of emotional distress on sleep patterns and whether it might explain the unhealthy sleep disparity (i.e., short or long sleep, irregular or interrupted sleep) between immigrants and non-immigrants. Emotional distress is defined as a state of emotional suffering characterized by symptoms of depression and anxiety [14], which can affect the overall health and well-being of an individual [15]. In fact, the often unrecognized and under-analyzed relationship between emotional distress and socio-demographic factors may obscure an independent relationship between emotional distress and sleep. 

Based on the literature, we hypothesized that immigrants suffer more from emotional distress and experience more insomnia symptoms than the non-immigrant population. Furthermore, we hypothesized that emotional distress would partially account for the relationship between immigrant status and perceived sleep quality and insomnia symptoms.

## 2. Materials and Methods

### 2.1. Participants

The Swiss Health Survey (SHS) is a recurring survey that investigates health status, socio-medical consequences of disease, and health care utilization in a representative sample of the resident population of Switzerland, including Swiss citizens and foreign nationals with a visa status valid for at least 12 months [16]. Participants in the survey are selected based on registries of inhabitants using a stratified random sampling technique, and are at least 15 years of age. To date, the survey has been conducted five times, in 1992/1993, 1997 and 2002, 2007, and 2012; the present study examines data collected in the most recent survey. 

The 2012 survey contacted 41,008 addresses, of which 21,597 people living in private households in Switzerland agreed to be interviewed by telephone (return rate: 53.1%). These participants then received a written questionnaire in German, French or Italian. The return rate of the questionnaire was 87.9%, which corresponded to 18,357 responders, for an overall participation rate of 44.7%. The data regarding sleep, but also sociodemographic data, stems from both the telephone and written interviews. Individuals who did not speak any of these three languages were included in the survey (proxy interview, *n* = 3.3%) if they had a person who translated for them. Following the sampling, data was weighted according to age, gender, marital status, citizenship, the number of family members, and region of residence to achieve a representative result [16].

### 2.2. Measures

The quality of sleep and insomnia symptoms were coded using four questions, assessing trouble falling asleep, trouble staying asleep, early morning awakening, and restless sleep. The questions were rated on a Likert scale (1 = often; 2 = sometimes; 3 = rarely; 4 = never). A total score for sleep disturbance was calculated as the mean of the four ratings, then a dichotomized value for the presence of symptoms was defined using the values ‘often’ and ‘sometimes’ (cutoff value: >2 versus ≤2). For the graphic depiction, we used inverted values, with four being the highest possible value.

Emotional distress (ED) was measured using the five-question Mental Health Inventory (MHI-5). There were six possible answers to the questions, scored between 1 and 6. The score for each individual therefore ranged between 5 and 30 and was then linearly transformed into a variable ranging from 0–100, which is a standard procedure for this instrument [17]. For easier comprehension, the data was recoded so that high values represent high levels of distress.

### 2.3. Co-Variates 

All analyses were adjusted for socio-demographic factors (marital status and household income), health risk behaviors (smoking and alcoholic drinking), and chronic diseases (hypertension diagnosis, heart problem, cancer diagnosis and diabetes diagnosis). Other covariates included age, body mass index (BMI), gender, and having children under the age of 15. All covariates were based on self-reporting during the phone and written interviews of the Swiss Health Survey.

### 2.4. Data Analysis 

We used the statistical software IBM SPSS 24.0 [18]. We constructed a descriptive table using Chi-square analysis for population comparison (immigrants vs. non-immigrants). We used a generalized linear model to determine whether emotional distress and immigration status were independently associated with insomnia symptoms. Four models were evaluated in our analyses. Model 1 was unadjusted with and without interaction, model 2 was age- and sex-adjusted, model 3 was adjusted for other patient characteristics (such as BMI, marital status, and SES), and model 4 was adjusted for medical comorbidities. The interaction between ED and sleep disturbance was coded so that the interaction effect reflected the increase in the likelihood of insomnia symptoms. For a more detailed description of the model construction please refer to Seixas, et al. [19].

## 3. Results

### 3.1. Descriptive Statistics 

Of the total participants, 18.6% were immigrants, of which more than half of the respondents were women (see Table 1). Compared to the non-immigrant population, immigrants were younger, more often married, and had a lower household income. Immigrants endured higher levels of emotional distress, had a higher body mass index, and suffered less frequently from arterial hypertension (Table 1). 

Table 2 presents the countries of origins of the immigrant population. A majority of people had arrived from other European countries, mainly Germany, Italy, France, and Portugal. Over half of the immigrant population had been living in Switzerland for 20 years or longer.

Of the total sample, 24.7% suffered from moderately to severely disturbed sleep based on experiencing sleep disorder symptoms sometimes or often. Figure 1 presents the differences between the two groups, showing that immigrants suffer more often from trouble falling asleep (ρ_s_ = −0.07, *p* < 0.001), disrupted sleep (ρ_s_ = −0.03, *p* = 0.001), and early awakening (ρ_s_ = −0.07, *p* < 0.001) than non-immigrants. However, restless sleep did not exhibit any significant difference between the two groups. 

Figure 2 presents a stratified analysis of sleep disorders showing that immigrants from Asia and Africa endorsed more sleep disorders in the first years after arrival to the host country with a subsequent constant decline. European and American immigrants, on the other hand, had a constant level of insomnia symptoms across the years since arrival.

### 3.2. Inferential Statistics 

Based on the ordinal logistic regression analysis, in the unadjusted model immigrants have 20.9% increased odds of reporting some symptom of disturbed sleep compared to non-immigrants (Table 3). Emotionally distressed individuals were 3.7 times more likely to report insomnia symptoms. Most importantly, there appears to be a significant interaction effect between immigrant status and ED on sleep. The results indicate that immigrants with higher ED scores had a 9.6% higher prevalence of insomnia symptoms in the unadjusted and the adjusted models compared to the Swiss native population (Table 3). The effects of ED and immigrant status persisted even when stratified by sex. However, immigrant status and ED interaction only continued to be significant in the male group (Table 4 and Table 5).

## 4. Discussion

The current study explored whether emotional distress might explain differences in insomnia symptoms between immigrants and non-immigrants. We found that immigrants had higher rates of insomnia symptoms compared to non-immigrants, which is consistent with previous findings [9,10]. 

Our finding that immigrants reported greater levels of emotional/psychological distress compared to non-immigrants is consistent with previous studies [20]. These studies illustrate that distress among immigrants is more likely if the individual is unemployed, living below the at-poverty-risk-threshold, and is of a lower socioeconomic status. Other studies indicate that social strain, such as excessive demands, overprotection, and rejection, may be related to emotional distress [21]. As evidenced in the current study, emotional distress corrected for socio-demographic, behavioral, and chronic disease factors was predictive of insomnia symptoms for both immigrants and non-immigrants. However, immigrants compared to non-immigrants had a higher prevalence of emotional distress, insomnia symptoms, and appeared to be more vulnerable to the ill effects of emotional distress on sleep.

In comparison to Swiss natives, higher levels of emotional distress among immigrants were associated with increased symptoms of insomnia in the male population. These findings suggest that male immigrants’ sleep is more sensitive to higher levels of emotional distress compared to non-immigrants. The lack of interaction in the female population is unexpected. We hypothesize that female immigrants might be less susceptible to experiencing ED-related sleep disturbance due to experiencing different sources of ED relative to men. Aichberger et al. [20] found in their study that socioeconomic status was only related to emotional distress in female Turkish immigrants without a partner. The authors hypothesized that difficulties may emerge when immigrant women from more collectivistic and family-oriented cultures try to be economically and socially independent. Our findings might reflect a reality of immigrant women with traditional family values leading to reduced contact with the host culture. Due to classical gender roles, females tend to care for the families and stay at home, thus showing a reduced integration into the host society, while males are forced to integrate due to their work environment [22]. Previous studies attribute sleep differences between immigrants and native populations to acculturation phenomena. Immigrants are likely to adopt negative health behaviors and increased stress as they adjust to living in the host country [7]. The process of societal integration appears to have negative consequences on sleep quality [10]. 

Past studies have indicated that biological, socioeconomic, health, medical comorbidities, and psychosocial factors may explain differences in sleep between immigrants and non-immigrants. Firstly, studies have shown that immigrants face various chronic stressors, which might lead to somatic stress responses. Physiological reactions may have adverse effects on sleep and in fact may partly explain the difference between immigrants and non-immigrants [23,24]. Secondly, studies have shown that acculturation in the host country might have an impact on the physical health status of immigrants [25]. Chronic illnesses such as obesity, diabetes, and hypertension can in return affect sleep quality [26]. Thirdly, psychosocial factors like socioeconomic status (low education and unemployment), low income, and being unmarried are related to sleep disruption [27]. Immigrants in Switzerland experience more difficulties in finding jobs and integrating into the community compared to non-immigrants [28].

One other possible explanation for the sleep disparity between immigrants and non-immigrants may be due to discrimination-related stress, since immigrants disproportionately encounter higher levels of prejudice and discrimination compared to non-immigrants [28]. Fischer et al. [23] found that immigrants reported higher levels of perceived ethnic discrimination, which resulted in increased stress. We argue that chronic exposure to difficulties in the acculturation process and discrimination may lead to emotional distress among immigrants, which in turn may induce a stress response and then a disruption in sleep.

Despite the plethora of evidence mentioned above for the determinants of sleep disruptions, to our knowledge, very few studies have found which determinants significantly lead to sleep differences between immigrants and non-immigrants. 

### Limitations and Future Directions

The current paper’s findings should be interpreted cautiously in light of several potential limitations. Firstly, the accepted languages were German, French, and Italian, which might have selected only those participants who have been assimilated to a certain degree to the host culture. Secondly, the use of self-reported sleep disorder symptoms, as opposed to objective measurement of sleep from polysomnography, may not yield the most accurate measurement of an individual’s normal sleep. Thirdly, the cross-sectional design of the SHS data prohibits us from making causal inferences between emotional distress and sleep. Lastly, we were unable to determine the nature and source of emotional distress, which would affect our interpretation of how emotional distress affects sleep. Future studies should investigate whether significant reductions in emotional distress might improve overall sleep quality among immigrants and therefore reduce sleep disparity between immigrants and non-immigrants. Additionally, future studies should include other emotionally distressing symptoms beyond symptoms of depression and anxiety. 

## 5. Conclusions

Despite these limitations, our study adds significantly to the literature by suggesting that emotional distress may explain sleep disparities between immigrants and non-immigrants in a Swiss national data set. Sleep disorders are a significant public health problem in Switzerland and all over the world, especially among immigrants, who are disproportionately affected. While sleep is profoundly influenced by distal biological, environmental, health, and psychosocial factors that are hard to modify, our finding that emotional distress, a more proximal and modifiable factor, is linked to sleep disorder symptoms might explain sleep disparities between immigrants and non-immigrants. 

## Figures and Tables

**Figure 1 ijerph-16-00289-f001:**
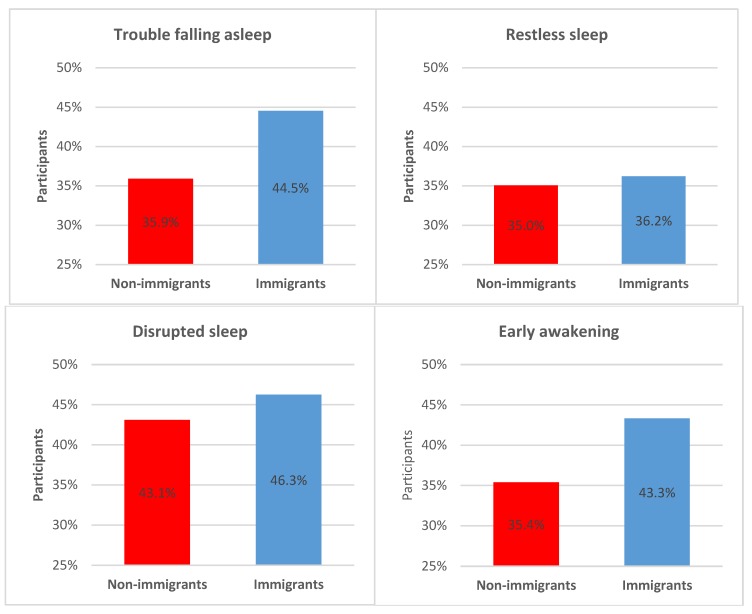
Sleep disorder symptoms stratified by migration status. Total population *N* = 17,968, non-immigrant population *n* = 14,562, immigrant population *n* = 3406. Multiple answers possible: trouble falling asleep, *p* < 0.001; restless sleep, *p* = 0.194; disrupted sleep, *p* = 0.001; early awakening, *p* < 0.001.

**Figure 2 ijerph-16-00289-f002:**
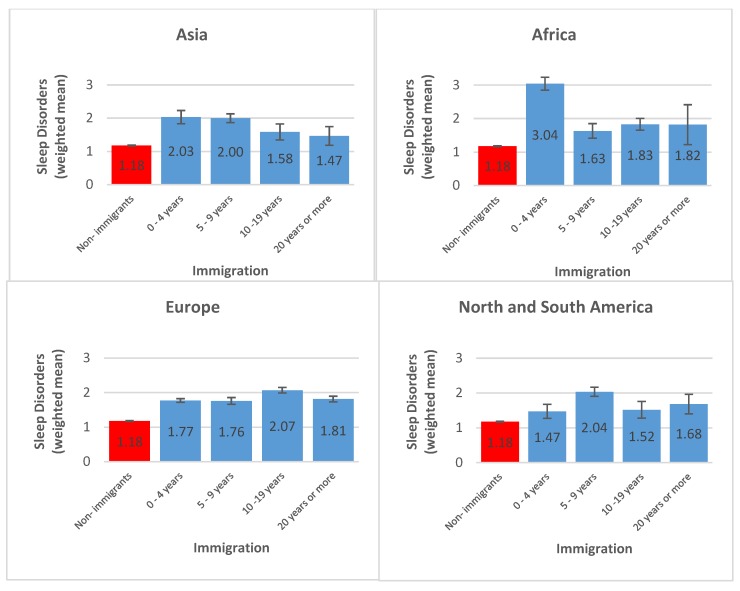
Sleep disorders stratified by continent of origin and years since immigration. Total population *N* = 17,968, immigrant population *n* = 3406, non-immigrant population *n* = 14,562, Europe *n* = 2927, North and South America *n* = 231, Africa *n* = 167, Asia *n* = 75, Australia and Oceania were omitted due to *n* = 6.

**Table 1 ijerph-16-00289-t001:** Characteristics of study participants by immigration status.

Variables	Immigrants		Non-Immigrants		
Sociodemographic variables	*n*		*n*		*p*-value
Gender (%)					0.540
Females	1815	(53.3%)	7675	(52.7%)	
Males	1591	(46.7%)	6887	(47.3%)	
Age (SD)	47.6	(15.8)	48.6	(18.8)	0.001
Marital status (%)					<0.001
Single	914	(22.1%)	5257	(32.4%)	
Married	2583	(62.4%)	8465	(52.1%)	
Registered partners	12	(0.3%)	37	(0.2%)	
Separated	77	(1.9%)	165	(1.0%)	
Divorced	386	(9.3%)	1313	(8.1%)	
Widowed	167	(5.7%)	999	(6.1%)	
Others	0	(0.0%)	3	(0.0%)	
Monthly household income Swiss francs (SD)	4752	(6400)	4981	(6425)	<0.001
Clinical variables	Immigrants		Non-immigrants		
Health variables					
Emotional distress rating (SD)	19.4	(16.7)	15.5	(13.9)	<0.001
BMI kg/m^2^ (SD)	24.8	(4.8)	24.3	(5.0)	<0.001
	*n*		*n*		*p*-value
Arterial hypertension (%)	466	(13.3)	2391	(16.4)	<0.001
Heart disease (%)	48	(1.4)	260	(1.8)	0.128
Cancer (%)	93	(2.9)	424	(2.9)	0.569
Diabetes mellitus (%)	150	(4.4)	653	(4.5)	0.838

Note: Total population: *N* = 17,968, immigrants: *n* = 3406, non-immigrants *n* = 14,562, Emotional distress rating: 1 (min. value)–100 (max. value).

**Table 2 ijerph-16-00289-t002:** Continents of origin and years since immigration.

Continent of Origin	Females		Males	
	*n*	%	*n*	%
Continent				
Europe	1531	84.35%	1396	87.74%
North and South America	156	8.60%	75	4.71%
Africa	84	4.63%	83	5.22%
Asia	40	2.20%	35	2.20%
Australia and Oceania	4	0.22%	2	0.13%
Years since immigration				
20 years or more	950	52.34%	831	52.23%
10–19 years	399	21.98%	294	18.48%
5–9 years	240	13.22%	247	15.52%
0–4 years	226	12.45%	219	13.76%

Note: Total population *N* = 17,968, immigrant population *n* = 3406, female immigrant population *n* = 1815, male immigrant population *n* = 1591.

**Table 3 ijerph-16-00289-t003:** Ordinal logistic regression models: associations between immigration status, emotional distress, and insomnia symptoms (total population).

	Variables	OR	95% CI
Model 1	Immigrant status	1.209	[1.109, 1.319] *
	Emotional distress	3.718	[3.431, 4.028] *
Model 1a	Immigrant status	1.208	[1.103, 1.323] *
	Emotional distress	3.577	[3.267, 3.918] *
	Immigrant status × emotional distress	1.096	[1.040, 1.154] **
Model 2	Immigrant status	1.196	[1.091, 1.312] *
	Emotional distress	3.608	[3.294, 3.952] *
	Immigrant status × emotional distress	1.097	[1.041, 1.155] *
Model 3	Immigrant status	1.255	[1.143, 1.377] *
	Emotional distress	3.522	[3.214, 3.859] *
	Immigrant status × emotional distress	1.093	[1.039, 1.151] **
Model 4	Immigrant status	1.277	[1.163, 1.402] *
	Emotional distress	3.478	[3.173, 3.812] *
	Immigrant status × emotional distress	1.096	[1.041, 1.153] *

Model 1: unadjusted; Model 1a: unadjusted with interaction; Model 2: adjusted for age and sex (weighted); Model 3: adjusted for patient characteristics (model 2 covariates plus weighted covariates: body mass index, marital status, income, children, smoking, alcohol consumption); Model 4: adjusted for comorbidity (model 3 covariates plus weighted covariates: arterial hypertension, heart disease, cancer, diabetes mellitus); OR: odds ratio; 95% CI: 95% confidence interval; * *p* < 0.001, ** *p* < 0.05.

**Table 4 ijerph-16-00289-t004:** Ordinal logistic regression models: associations between immigration status, emotional distress, and insomnia symptoms (women).

	Variables	OR	95% CI
Model 1	Immigrant status	1.181	[1.052, 1.325] *
	Emotional distress	3.606	[3.251, 4.001] **
Model 1a	Immigrant status	1.181	[1.049, 1.328] *
	Emotional distress	3.557	[3.186, 3.972] **
	Immigrant status × emotional distress	1.045	[0.978, 1.116]
Model 2	Immigrant status	1.144	[1.016, 1.289] *
	Emotional distress	3.576	[3.202, 3.993] **
	Immigrant status × emotional distress	1.046	[0.979, 1.117]
Model 3	Immigrant status	1.210	[1.073, 1.366] *
	Emotional distress	3.524	[3.155, 3.936] **
	Immigrant status × emotional distress	1.040	[0.974, 1.111]
Model 4	Immigrant status	1.229	[1.089, 1.387] *
	Emotional distress	3.524	[3.155, 3.937] **
	Immigrant status × emotional distress	1.041	[0.974, 1.111]

Model 1: unadjusted; Model 1a: unadjusted with interaction; Model 2: adjusted for age (weighted); Model 3: adjusted for patient characteristics (model 2 covariates plus weighted covariates: body mass index, marital status, income, children, smoking, alcohol consumption); Model 4: adjusted for comorbidity (model 3 covariates plus weighted covariates: arterial hypertension, heart disease, cancer, diabetes mellitus); OR: odds ratio; 95% CI: 95% confidence interval; * *p* < 0.05, ** *p* < 0.001.

**Table 5 ijerph-16-00289-t005:** Ordinal logistic regression models: associations between immigration status, emotional distress, and insomnia symptoms (men).

	Variables	OR	95% CI
Model 1	Immigrant status	1.252	[1.098, 1.429] *
	Emotional distress	3.660	[3.222, 4.157] **
Model 1a	Immigrant status	1.246	[1.082, 1.434] **
	Emotional distress	3.325	[2.832, 3.903] **
	Immigrant status × emotional distress	1.169	[1.075, 1.271] *
Model 2	Immigrant status	1.229	[1.065, 1.418] *
	Emotional distress	3.320	[2.826, 3.899] **
	Immigrant status × emotional distress	1.172	[1.078, 1.274] **
Model 3	Immigrant status	1.319	[1.141, 1.525] **
	Emotional distress	3.326	[2.828, 3.911] **
	Immigrant status × emotional distress	1.175	[1.081, 1.276] **
Model 4	Immigrant status	1.346	[1.164, 1.556] **
	Emotional distress	3.285	[2.794, 3.861] **
	Immigrant status × emotional distress	1.169	[1.077, 1.269] **

Model 1: unadjusted; Model 1a: unadjusted with interaction; Model 2: adjusted for age (weighted); Model 3: adjusted for patient characteristics (model 2 covariates plus weighted covariates: body mass index, marital status, income, children, smoking, alcohol consumption); Model 4: adjusted for comorbidity (model 3 covariates plus weighted covariates: arterial hypertension, heart disease, cancer, diabetes mellitus); OR: odds ratio; 95% CI: 95% confidence interval; * *p* < 0.05, ** *p* < 0.001.

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
