# Peer review of "Differences in Insomnia Symptoms between Immigrants and Non-Immigrants in Switzerland attributed to Emotional Distress: Analysis of the Swiss Health Survey"

_ijerph, 2019, doi:10.3390/ijerph16020289_

Round 1
Reviewer 1 Report
This paper is interesting and addresses a population of people whose numbers are growing on the planet. It would have been even nicer if the survey would have stratified refugee population as a subset of all migrants.
Line 45-46: Minor grammar issues are apparent in line 45-46: Since this statement is based on data from 2014, past tense should be used. Sentences beginning with a number (e.g. 80%) should be spelt out (Eighty percent); also, were immigrants (not are immigrants); and, one-fifth were born (not was born)
Line 126: Table 1 on p4 should state the currency used to measure income (is it Euros?)
Line 126: Table 1 needs to be reformatted on p4 where Emotional Distress (SD), BMI (SD) are listed. Here the numbers do not denote 'n' (population count). These variable are measured differently (e.g. BMI kg/m2 ) and should be listed as a separate section in the table. The heading of 'n' correctly applies to the other variables (Arterial hypertension, Heart disease, Cancer and Diabetes mellitus)
Line 132: The second sentence should begin 'A majority.....' (not The majority).
Line 146 Figure 2: All the 4 sub-figures should be reformatted to show the immigration bar graphs in chronological order starting with the 0-4 years on the left and marching to 20 years or more to the right. This is how the text reads in lines 152-155 and this is how the reader would most easily imagine seeing the progression of changes in the sub-populations in the graphics. The current presentation is backwards.
Line 215-217 should be placed after lines 218-224. By making this switch the flow of the discussion from line 205- 214 will not be interrupted and all possible explanations will be dealt with together.
Thank you.
Author Response
This paper is interesting and addresses a population of people whose numbers are growing on the planet. It would have been even nicer if the survey would have stratified refugee population as a subset of all migrants.
Thank you very much for the review. We agree that this is a growing issue and that most importatnly refugee populations are at risk. Unfortunately, the Swiss Health Survey did not differentiate refugee populations from other migrants.
Line 45-46: Minor grammar issues are apparent in line 45-46: Since this statement is based on data from 2014, past tense should be used. Sentences beginning with a number (e.g. 80%) should be spelt out (Eighty percent); also, were immigrants (not are immigrants); and, one-fifth were born (not was born)
Thank you. We corrected accordingly.
Line 126: Table 1 on p4 should state the currency used to measure income (is it Euros?)
It is Swiss Francs, we added it to the Table.
Line 126: Table 1 needs to be reformatted on p4 where Emotional Distress (SD), BMI (SD) are listed. Here the numbers do not denote 'n' (population count). These variable are measured differently (e.g. BMI kg/m2 ) and should be listed as a separate section in the table. The heading of 'n' correctly applies to the other variables (Arterial hypertension, Heart disease, Cancer and Diabetes mellitus)
Thank you for reviewing our manuscript attentively. We adjusted the table accordingly.
Line 132: The second sentence should begin 'A majority.....' (not The majority).
Corrected
Line 146 Figure 2: All the 4 sub-figures should be reformatted to show the immigration bar graphs in chronological order starting with the 0-4 years on the left and marching to 20 years or more to the right. This is how the text reads in lines 152-155 and this is how the reader would most easily imagine seeing the progression of changes in the sub-populations in the graphics. The current presentation is backwards.
This is true. We changed the figure accordingly.
Line 215-217 should be placed after lines 218-224. By making this switch the flow of the discussion from line 205- 214 will not be interrupted and all possible explanations will be dealt with together.
We made the change as recommended.
Thank you.
Reviewer 2 Report
A fairly will return article with low importance, may be selecting a broader population might have been helpful, specifying the cultural background off the immigrants may have been helpful as well. We have a lot of information about immigrants in United States which becomes a baseline for your study, and we are also aware about the sleep disturbance is caused by excessive stress. This is seen in all immigrant population and native population and does not provide much importance but will return article overall.
Author Response
A fairly will return article with low importance, may be selecting a broader population might have been helpful, specifying the cultural background off the immigrants may have been helpful as well. We have a lot of information about immigrants in United States which becomes a baseline for your study, and we are also aware about the sleep disturbance is caused by excessive stress. This is seen in all immigrant population and native population and does not provide much importance but will return article Overall.
Thank you for your comments. It is true that most studies regarding migrant populations, stress and sleep are based in the US. Therefore, we think that studies from other countries are essential to expand the existing data. Using the data from National Health Surveys limits the specification we can make about the different cultural backgrounds. However, we are convinced that the manuscript still adds to the scientific knowledge base and fills the void of data regarding migrants in other countries than the US.
Reviewer 3 Report
This manuscript highlighted an important issue, i.e. what may cause the increased sleep disruptions and impaired sleep quality among immigrants compared to the native residents in a developed country like Switzerland. The authors suggested that emotional distress mediates in between immigration experience and sleep disruptions. The study is well conceived and written, and the authors should be commended for bringing up this topic. However I have a few questions and suggestions pending further responses.
1. Introduction, ref 1: shouldn’t it be 2.44 billion if it weighs 32.9% of the world’s population?
2. The authors tried to assume that immigrants were more frequently exposed under stress and thus might suffer increased sleep problems, however the Mexican immigrants in the US here did not support such an assumption. Please find other proper evidence here to replace the existing one.
3. 2.2 measure of sleep disturbances: What was the cutoff value? Was it >8 vs =<8 for total score, please clarify.
4. Line 104-107, please use past tense.
5. Line 106-107, was this transformation into 0-100 a standard procedure in the analysis of MHI-5? If not, why?
6. Covariates - chronic diseases: were they based on self-report cases or extracted from medical history?
7. I suggest using insomnia symptoms instead of disturbed sleep throughout the manuscript as all four questions regarding sleep quality were about insomnia symptoms (neither sleep duration nor any other sleep disorders).
Author Response
This manuscript highlighted an important issue, i.e. what may cause the increased sleep disruptions and impaired sleep quality among immigrants compared to the native residents in a developed country like Switzerland. The authors suggested that emotional distress mediates in between immigration experience and sleep disruptions. The study is well conceived and written, and the authors should be commended for bringing up this topic. However I have a few questions and suggestions pending further responses.
Thank you very much for your comments and review. It helped us improve our manuscript.
1. Introduction, ref 1: shouldn’t it be 2.44 billion if it weighs 32.9% of the world’s population?
This is correct, we changed it accordingly.
2. The authors tried to assume that immigrants were more frequently exposed under stress and thus might suffer increased sleep problems, however the Mexican immigrants in the US here did not support such an assumption. Please find other proper evidence here to replace the existing one.
We removed the reference and added one that supports our research question:
Maneze, D., Salamonson, Y., Poudel, C., DiGiacomo, M., Everett, B., & Davidson, P. M. (2016). Health-seeking behaviors of Filipino migrants in Australia: The influence of persisting acculturative stress and depression. Journal of immigrant and minority health, 18(4), 779-786.
3. 2.2 measure of sleep disturbances: What was the cutoff value? Was it >8 vs =<8 for total score, please clarify.
Thank you for pointing this out. We rephrased the text including the cutoff value. The text reads as follows:
The questions were rated on a Likert scale (1=often; 2=sometimes; 3=rarely; 4=never). A total score for sleep disturbance was calculated as the mean of the four ratings, then a dichotomized value for presence of symptoms was defined using the values often and sometimes (cutoff value: >2 versus =<2). For the graphic depiction, we used inverted values with four being the highest possible value.
4. Line 104-107, please use past tense.
We changed the text accordingly.
5. Line 106-107, was this transformation into 0-100 a standard procedure in the analysis of MHI-5? If not, why?
Yes, it is a Standard procedure. We added this to the text for more clarity.
6. Covariates - chronic diseases: were they based on self-report cases or extracted from medical history?
They were based on self-Report. We added the following sentence to the text:
All covariates were based on self-report during the phone and written interview of the Swiss Health Survey
7. I suggest using insomnia symptoms instead of disturbed sleep throughout the manuscript as all four questions regarding sleep quality were about insomnia symptoms (neither sleep duration nor any other sleep disorders).
Thank you for being attentive to the details of our manuscript. We changed the wording and replaced disturbed sleep with insomnia symptoms.